# Opinion: Aerosol Remote Sensing Over The Next Twenty Years

Lorraine A. Remer[1], Robert C. Levy[2], J. Vanderlei Martins[3]

[1]Goddard Earth Sciences Technology and Research 2 (GESTAR 2), University of Maryland Baltimore County, Baltimore MD, 21228, USA

[2]NASA Goddard Space Flight Center, Greenbelt MD, 20771, USA

[3]Department of Physics, University of Maryland Baltimore County, Baltimore MD, 21250, USA

*Correspondence to*: Lorraine A. Remer (remer@umbc.edu)

**Abstract.** More than two decades ago, aerosol remote sensing underwent a revolution with the launch of the Terra and Aqua satellites. Advancement continued via additional launches carrying new passive and active sensors. Capable of retrieving parameters characterizing aerosol loading, rudimentary particle properties and in some cases aerosol layer height, the satellite view of Earth's aerosol system came into focus. The modeling communities have made similar advances. Now the efforts have continued long enough that we can see developing trends in both remote sensing and modeling communities, allowing us to speculate about the future and how the community will approach aerosol remote sensing twenty years from now. We anticipate technology that will replace today's standard multi-wavelength radiometers with hyperspectral and/or polarimetry all viewing in multiple angles. These will be supported by advanced active sensors with the ability to measure profiles of aerosol extinction in addition to backscatter. The result will be greater insight into aerosol particle properties. Algorithms will move from being primarily physically-based to include an increasing degree of Machine Learning methods, but physically-based techniques will not go extinct. However, the practice of applying algorithms to a single sensor will be in decline. Retrieval algorithms will encompass multiple sensors and all available ground measurements into a unifying framework, and these inverted products will be ingested directly into assimilation systems, becoming "cyborgs": half observations, half model. In twenty years we will see a true democratization in space with nations large and small, private organizations and commercial entities of all sizes launching space sensors. With this increasing amount of data and aerosol products available, there will be a lot of bad data. User communities will organize to set standards and the large national space agencies will lead the effort to maintain quality by deploying and maintaining validation ground networks and focused field experiments. Through it all, interest will remain high in the global aerosol system and how that system affects climate, clouds, precipitation and dynamics, air quality, the environment and public health, transport of pathogens and fertilization of ecosystems, and how these processes are adapting to a changing climate.

## 1 NASA initiates a revolution with the Terra and Aqua missions

As NASA's Terra and Aqua satellites launched at the end of 1999 and in mid-2002, respectively, they ushered in a new era of aerosol remote sensing. These satellites carrying the MODerate resolution

Imaging Spectroradiometer (MODIS) (Salomonson et al., 1989) and the Multiangle Imaging SpectroRadiometer (MISR, on Terra only)(Diner et al., 1998) offered an unprecedented view of the global aerosol system (Kaufman et al., 2002). The products derived from MODIS and MISR observations defined an aerosol climatology (Remer et al., 2008, Voss and Evan, 2020), identified

regional trends (Yu et al., 2020, Zhang and Reid, 2010; Hammer et al., 2020), advanced understanding of aerosol direct radiative forcing (Christopher and Zhang, 2002, Zhang and Christopher, 2003, Yu et al., 2006, Remer and Kaufman, 2006, Bellouin et al. 2008, Li et al., 2016), aerosol indirect radiative forcing (Peng et al., 2016)), discovered associations between aerosols, clouds and precipitation that shifted our perception of the water cycle (Kaufman et al., 2005, Koren et al., 2005, 2008, Yuan et al.,

2011ab, Niu and Li, 2012), proved the importance of aerosol intercontinental transport and nutrient deposition (Gao et al., 2001, Yu et al., 2013, 2019), and gave us important new tools to determine aerosol plume heights (Kahn et al., 2007, 2008, Val Martin et al., 2018) and air quality monitoring and mitigation (Engel-Cox, 2004, Van Donkelaar et al., 2006, 2016, Gupta and Christopher, 2009ab).

During this period, aerosol remote sensing progressed as a discipline. The MODIS and MISR algorithms had to run in an operational environment, which meant they had to work for all types of aerosols, and in all types of environments. The MODIS and MISR algorithms are rooted in the physics of radiative transfer and the mathematics of inversions, but they also incorporate empirical assumptions that tie results to the real world (Kaufman et al., 1997, Tanré et al. 1997, Martonchik et al., 1998, 2009,

Kahn et al., 2001). Assumptions are necessary because of the fundamental lack of information content in the MODIS and MISR observations (Tanré et al. 1996, Kahn et al., 1998). MODIS provides a multi-spectral look at each Earth scene at a single viewing geometry. The original MODIS aerosol algorithms, Dark Target (Remer et al., 2005) and Deep Blue (Hsu et al. 2006), make use of no more than six wavelengths for their retrieval, although they use additional wavelengths for cloud masking and other

peripheral needs. The MODIS Dark Target over ocean retrieval makes use of the most spectral bands, six. These six bands contain only 2-3 pieces of independent aerosol information (Tanré et al., 1996). Over land, fewer bands are used and the information content available to characterize aerosol is even less Levy et al., 2007). MISR employs fewer wavelengths, only four from 0.486 µm to 0.867 µm, but adds significant additional information by viewing each scene from 9 different angles. MISR's angular

information has shown to be immensely valuable in constraining aerosol type and in determining aerosol plume height (Kahn et al., 2007, 2008, Val Martin et al., 2018, Kahn and Gaitley, 2015).

However, in addition to the advances leading to the global satellite retrievals in an operational setting, these products also had to be validated and interpreted. The development of MODIS, MISR, and then

the active sensors led to significant advances in ground-based remote sensing (e.g AERONET; Holben

et al., 1998 for validation and algorithm improvements). At the same time, they also led to advances in the modeling and assimilation communities.  By providing accurate global products across all types of environments, now models have (a) something to compare with, and (b) something to assimilate.

MODIS and MISR created a revolution in aerosol remote sensing when they were launched in 1999, but the reasons for the revolution involved more than new satellite technology married to state-of-the-science algorithms.  A combination of factors came together at just the right time.  These factors include:

- A policy for open data access,
- An investment in infrastructure for processing algorithms and distributing data,
- A commitment to long-term science teams for maintenance, improvement and validation of data products including on-orbit sensor calibration,
- The vision and continuous support of the AERONET program for product validation (Holben et al., 1998),
- The overall growth of the international aerosol community and expanded interest in the global aerosol system as part of Earth system science.

We find the first bullet point of particular interest for the celebration of the twentieth anniversary of the journal *Atmospheric Chemistry and Physics*, which itself propelled a revolution in open access and
information sharing at roughly the same time.

Thus, the lessons learned from the previous 20 years in aerosol remote sensing is that technical innovation leads to more observable information that leads to more complete aerosol characterization at higher accuracy. But that chain of advancement will only seed new scientific endeavors if the aerosol
products are accessible, validated, accompanied by documentation that a user can actually understand and part of a larger data effort to characterize not only the global aerosol system but the Earth system as a whole. The infrastructure created for MODIS and MISR (LAADS DAAC, 2023, ASDC, 2023) continued to absorb and produce data products derived from other sensors and missions over the past two decades: the Aura mission, Cloud-Aerosol Lidar with Orthogonal Polarization (CALIOP) data from
Cloud-Aerosol Lidar and Infrared Pathfinder Satellite Observations(CALIPSO), the Polarization and Anisotropy of Reflectances for Atmospheric Sciences coupled with Observations from a Lidar (PARASOL) mission.  Other agencies adopted NASA's open access policies for their missions and products:  Visible Infrared Imaging Radiometer Suite (VIIRS), Himawari, Sentinels, Geostationary Ocean Color Instrument (GOCI ) etc.  Today, a user needing local or global aerosol characterization
may feel overwhelmed with the number of aerosol products available to use, both those derived directly from satellite observations and those inferred from models that use satellite information in assimilation systems (Zhang et al., 2008, Benedetti et al., 2009, Gelaro et al., 2017). The perspective from 2023 is

that of an apparently data rich era that could hardly have been imagined 23 years ago when Terra launched**.**


*But what will aerosol remote sensing and data products look like over the next 20 years?*


## 2 New technology leads to MORE, MORE, MORE information

The future of aerosol remote sensing will be propelled by advances in satellite payload technology. Instead of imaging multi-wavelength radiometers, like MODIS, we will expect the additional angular
information pioneered by MISR plus the advantages of polarization. Imaging multi-wavelength multi-angle polarimeters were introduced to the community with POLarization and Directionality of the Earth's Reflectances (POLDER)/PARASOL (Deuzé et al., 2000, 2001; Chen et al., 2020), Future imaging polarimeters will improve upon the POLDER/PARASOL technology with better polarization accuracy, hyper spectral and hyper angle capabilities, decreases in pixel size, increase in the number of
wavelengths with polarization and still allow for imaging. Super polarimeters with *all* of this technology in a single instrument are not necessary. Individual missions will tailor their instrument characteristics to meet particular mission needs, but these types of instrument advancement will be available for mission design.  The Multi-viewing, Multi-channel, Multi-polarisation imaging mission (3MI) is the European follow-on to POLDER/PARASOL that will fly on EUMETSAT's 2nd generation polar-
orbiting Meteorological Observing satellite series (MetOp SG) (Fougnie et al. 2018). The first of this series is planned for launch in 2024. The HyperAngle Rainbow Polarimeter 2 (HARP2) and the Spectro-polarimeter for Planetary Exploration One (SPEXone) will both fly on NASA's Plankton, Aerosol, Clouds, ocean Ecosystem (PACE) mission to be launched in 2024 (Werdell et al. 2019), and the Multi Angle Imager for Aerosols (MAIA: Diner et al., 2018) is scheduled for launch soon after.
China has the Directional Polarized Camera (DPC) already in orbit aboard the Daqi-1 (DQ-1) satellite, with similar sensors repeated to launch until 2033. Finally, India is planning an aerosol mission, Nanosatellite for Earth Monitoring and Observation-Aerosol Monitoring (NEMO-AM), that will carry a multiangle polarimeter the Multi Angle Dual Polarisation Imaging sensor (MADPI).

Similar instruments, with extended wavelength ranges, are planned for NASA's Atmospheric Observing System (AOS, 2023) launching later this decade.  Beyond AOS, we expect more hyperspectral capability, broader wavelength ranges including the thermal infrared, finer spatial resolution, broader swaths and more angles. Twenty years from now these imaging multi-angle and hyperspectral polarimeters will be the norm for passive aerosol characterizing sensors. Simple single-
view multi-wavelength radiometers will still be flying through the 2030s, as they offer products across multiple disciplines: Visible InfraRed Suite (VIIRS), Ocean and Land Colour Instrument (OLCI),

MultiSpectral Instrument (MSI), Medium Resolution Spectral Imager (MERSI), but the cutting edge aerosol characterization will have moved to the polarimeters by the end of the 2030s. The exception for radiometers will be the geostationary fleet of multispectral radiometers that we discuss below.


In the thermal infrared, single-view instruments will still have a role (DeSouza-Machado, 2010, Klüser et al. 2011). Hyperspectral thermal infrared spectrometers such as the Atmospheric Infrared Sounder (AIRS) have been used to detect and quantify dust AOD and layer height beginning with the Aqua mission (DeSouza-Machado et al., 2006) and advancing to other hyperspectral sounders in this spectral
range such as the Infrared Atmospheric Sounding Interferometer (IASI) (Clarisse et al., 2019). The community has recently directed greater focus to dust's contribution to Earth energy balance through its perturbation of the radiation field at infrared wavelengths (Zheng et al., 2022, Song et al., 2022). New technology for hyperspectral thermal infrared spectrometers will improve signal, reduce spatial footprints and are our best hope to achieve mineral speciation of airborne dust from space.


Along with advances in passive remote sensing imagers, we expect new capabilities in active lidars. The straightforward backscatter lidar in space (CALIOP) has demonstrated the unique aerosol information that only a lidar can provide. By 2043 standard aerosol missions will include lidars and these lidars will surpass CALIOP with additional wavelengths, High Spectral Resolution Lidar (HSRL) and/or Raman
capability, as well as multiple beams to expand coverage. Ground-based and airborne lidars with these enhanced capabilities are demonstrating that like polarimetry, information content from enhanced lidar design allows for aerosol characterization in ways previously unattainable (Müller et al., 2014). Post-CALIOP lidars lidars flying now or expected in the near future include: the Chinese Aerosol and Carbon Detection Lidar (ACDL; Ke et al., 2022)  and the European ATmospheric backscatter
LIDar (ATLID; Groß et al., 2015), both have  HSRL capability. These are small steps forward, and we understand the expense of launching high capability lidars in space, as the HSRL planned for NASA's AOS mission has recently been descoped because of cost. Yet, we remain optimistic. By 2043, these modest steps into lidar capability expansion could grow to achieve the ideal for aerosol characterization (Müller et al., 2014). This means that there could be at least one space payload offering 3 wavelengths
of backscattering, two wavelengths of extinction profiles using HSRL technology and one wavelength providing depolarization. Aerosol scientists have been advocating for this array of lidar payload capability for more than a decade, and despite the next set of planned aerosol missions falling short, we foresee the dream become reality in 20 years.

The lessons we learned from the past twenty years in terms of the advantages of formation flying will guide mission planning far into the future. Like the A-Train of the 2000s and 2010s, missions will be constructed to purposely match passive polarimeters with active lidars in ways that promote synthesized data processing, algorithms, science and applications. The workhorse sun-synchronous polar orbiting mission that has been the mainstay for aerosol characterization will be supplemented with inclined
orbits that have been more typical of precipitation-centered missions but have much to offer for aerosols. Inclined orbits allow for a multitude of geometries and time-of-day sampling of the aerosol system over the course of weeks as well as multiple coincidences a day with other satellites looking simultaneously at the same target at the ground. This provides a peek into aerosol situations never

encountered by MODIS, MISR or their ilk. Part of this expanding constellation of aerosol observations
from satellite will be the geostationary imagers (GEO).  Already capability from GEO platforms has
evolved from the two or three broad uncalibrated wavelengths that were ubiquitous at the time of Terra
launch into multi-wavelength moderate resolution sensors that mimic most of MODIS's capabilities. By
2043, these modern-day Advanced Baseline Imagers (ABI) and Advanced Himawari Imager (AHI),
FengYu (FY) and Meteosat Third Generation (MTG) will be upgraded with spectrometers, as per the
NASA/NOAA joint Geostationary Extended Observations (GEO-XO) program (Frost et al., 2020). At
that time we expect studies in progress to adapt polarimeters to GEO orbit. Other GEO missions include
the Geostationary Extended Observations Imager (GXI), Flexible Combined Imager (FCI),
Geostationary High speed Imager (GHI) and the Advanced Geostationary Radiation Imager (AGRI),
Lidar at that altitude may still be pending. With the equator crowded with observatories in GEO orbit,
overlap between sensors provides both a means to cross-calibrate and also multi-angle looks at each
scene to increase information content for aerosol retrieval (Bian et al., 2021).

Table 1 presents a partial list of planned space missions from the national agencies of U.S., Europe and
China that have identified an aerosol objective. The list is not comprehensive by any measure, as many
missions not specifically designed to characterize aerosol find a means to do so anyway. For example,
the Earth surface Mineral dust source Investigation (EMIT) mission will advance characterization of
dust aerosol by identifying mineral content of dust on the ground, before the particles are airborne.
EMIT will help with aerosol characterization but is not included here because it is not designed to
retrieve airborne aerosol parameters. Table 1 is also missing a comprehensive list of missions because
many additional countries are finding resources to launch space observatories as space becomes more
democratized, as we will discuss below. Already India, Japan and Korea are planning for future aerosol-
capable missions into the next decades.

Table 1. Partial list of planned missions from major space agencies in the U.S., China, Europe and one
each from India and South Korea having an aerosol objective or aerosol-capable without a specific
objective. Compiled from WMO OSCAR (2011-2023).

| Country | Mission<br>Orbit | Satellite (duration) | Sensor: type |
|---------|------------------|----------------------|--------------|
| U.S. | Joint Polar Satellite System (JPSS)<br><br>Sun synchronous | NOAA-21*, JPSS-3, -4 (2022-2030), (2027-2036), (2032-2039) | VIIRS: multispectral radiometer |
| U.S. | Plankton, Aerosol, Clouds, ocean Ecosystems (PACE)<br><br>Sun synchronous | PACE (2024 – 2028) | OCI: hyperspectral radiometer<br>HARP2 and SPEXone: multiangle imaging polarimeters |
| U.S. | Multi Angle Imager for Aerosols (MAIA) | MAIA (2024-2027) | MAIA: multiangle imaging polarimeter |

| | | Sun synchronous | |
|------|-------------------------------------------------------------------|--------------------------------------------------------------------|------------------------------------------------------------------------|
| U.S. | Atmosphere Observing System (AOS)  Sun synchronous | AOS-polar (2029-2039) | Multiangle imaging polarimeter;  Backscatter lidar |
| U.S. | Atmosphere Observing System (AOS)  Inclined | AOS-inclined  (2030-2040) | Backscatter lidar |
| U.S. | Geostationary Extended Observations (GeoXO)  Geostationary | GeoXO-East, GeoXO-West (2032-2047), 2035-2050) | GXI: multiangle radiometer |
| U.S. | Tropospheric Monitoring of Pollution (TEMPO) | TEMPO* (2023-2026) | Hyperspectral in GEO orbit |
| China | Gao-Fen (GF)-5 Sun synchronous | GF-01A*, 02* (2022-2030), (2021-2028), respectively | DPC: multiangle imaging polarimeter |
| China | Daqi (DQ) Sun synchronous | DQ-1*, DQ-2 (2022-2030), (2025-2033) | DPC: multiangle imaging polarimeter; ACDL: high spectral resolution lidar |
| China | FengYun (FY)-3 Sun synchronous | FY-3D*, 3F*, 3H (2012-2024), (2023-2032), (2024-2029), respectively | MERSI: multispectral radiometer |
| China | FengYun (FY)-4 Geostationary | FY-4B* (2021 – 2028) | GHI: multispectral radiometer |
| China | FengYun (FY)-4 Geostationary | FY-4C, 4D, 4E (2025-2031), (2026-2033), (2027-2034), respectively | AGRI: multispectral radiometer |
| | | | |
| Europe | Earthcare Sun synchronous | Earthcare (2024-2027) | ATLID: backscatter lidar MSI: multispectral radiometer |
| Europe | Sentinel-2 Sun synchronous | Sentinel-2C, 2D (2024-2031), | MSI: multispectral radiometer |

| | | (2025-2032) | |
|---|---|---|---|
| Europe | Sentinel-3<br>Sun synchronous | Sentinel-3C, 3D<br>(2024-2034),<br>(2028-2038) | OLCI: multispectral radiometer |
| Europe | EPS Second Generation<br>Sun synchronous | MetOp-SG-A1, A2, A3<br>(2025-2032),<br>(2032-2039)<br>(2039-2046) | 3MI: multiangle imaging polarimeter<br>METimage: multispectral radiometer |
| Europe | MeteoSat Third Generation (MTG)<br>Geostationary | MTG-I1*, I2, I3, I4<br>(2022-2030),<br>2026-2034),<br>(2032-2040),<br>(2036-2044) | FCI: multispectral radiometer |
| India | Nanosatellite for Earth Monitoring and Observation-Aerosol Monitoring (NEMO-AM)<br><br>Sun synchronous | NEMO-AM (2025-2029) | MADPI:<br>Multiangle imaging polarimeter |
| South Korea | Geostationary Korea MultiPurpose Satellite (G**EO**-KOMPSAT) | GEO-KOMPSAT-2A*<br>(2018-2029)<br>GEO-KOMPSAT-2B*<br>(2020-2031) | Advanced Meteorological Instrument (AMI): multispectral radiometer<br>Geostationary Environmental Monitoring Spectrometer (GEMS): Hyperspectral in GEO orbit<br>Geostationary Ocean Color Imager II (GOCI-II): multispectral radiometer |

- Already launched and in orbit

The tag line for technology in 2043 will be MORE, MORE, MORE!  Each sensor will be collecting orders of magnitude more data than does MODIS today. Polarized measurements require three times the data per wavelength than a simple radiometer to account for the three angles of polarization. Multiply that increase by the number of view angles beyond the single MODIS-like view and we are likely now producing 15X to 60X the amount of data being produced today. The implementation of hyperspectral sensors or improvements in pixel size while maintaining broad swaths to view the entire globe further

increases data volume. Advancing lidars with additional wavelengths, HSRL, multiple beams and quicker pulses also magnifies data volume quickly. As part of technology evolution will be the need for spacecraft and especially spacecraft electronics, data systems and transmission to keep up with data volumes. Optical transmission will replace radio frequencies and infrastructure in space will serve as
way stations to stage downlinking to overworked ground stations. For example, today the Ka-band is the radio frequency of choice for satellite missions. NASA is promoting "new and improved" Ka-band technology, available for licensing, that achieves a data rate of 130 megabits per second (130 Mbps) (NASA Technology Solution Communications, 2023). In parallel, NASA recently announced achieving 200 gigabytes per second (Gbps) with a space-to-ground optical link (NASA Press Release,
2023). That is three orders of magnitude more data coming down with an optical system. On Earth, computing power will be fully cloud-based and ever expanding to hold the proliferation of collected data.

## 3 Evolution of Algorithms

Today, the majority of aerosol retrieval algorithms interpret the signal measured by the satellite sensor in terms of radiative transfer through the atmosphere. The process is governed by this equation,

$$\rho_{TOA} = \rho_a + \frac{T\rho_s}{(1 - s\rho_s)}$$

Where $\rho_{TOA}$ is the solar radiance reflected by the Earth system measured at the top of the atmosphere by the satellite sensor, $\rho_a$ is the contribution by the atmosphere alone, $\rho_s$ is the reflectance of Earth's surface that transmits through the atmosphere with transmittance, $T$, and s is the spherical albedo with the $(1 - s\rho_s)$ term representing multiple scattering between the atmosphere and Earth's surface. Each term is for a specific wavelength, sun, and sensor geometries, but that notation has been omitted for
simplicity. The terms depend on surface properties ($\rho_s$ $and$ $s$) and the amount and optical properties of the gases and aerosols in the atmosphere ($\rho_a$ $and$ $T$). By making physically-based assumptions on some surface, gas and aerosol properties and calculating $\rho_{TOA}$, the calculated $\rho_{TOA}$ can be compared with the measured $\rho_{TOA}$ and assumptions adjusted until the calculated and measured $\rho_{TOA}$ are the same to within an acceptable error, consistent with measurement uncertainty.

The simplest physically-based algorithms use one wavelength, one geometry at a time. These simple retrievals hold all assumptions constant except for the optical measure of aerosol loading, the aerosol optical depth (AOD). Thus, the retrievals use one piece of information to retrieve one parameter, the AOD. By minimizing the difference between measured and calculated $\rho_{TOA}$ simultaneously for more

than one wavelength, or more than one angle, or more than one polarization state, more information is introduced and additional assumptions can become free parameters leading to retrievals of aerosol parameters beyond AOD. For example, the MODIS dark target retrieval is based on the knowledge that independent, multi-spectral observations provide more information. However, not all wavelengths are completely independent, so that seven MODIS wavelengths from 0.47 µm to 2.1 µm yield, at most, three pieces of information (Tanré et al., 1996). Likewise, additional geometries and/or polarization states can be introduced, freeing even more parameters for retrieval. The simplest algorithms have employed radiative transfer models run in the forward direction with assumed aerosol, gases, and surface parameters resulting in a Look Up Table (LUT) stored for future use. The retrieval then can move quickly through the LUT, interpolating between entries, and return the retrieved parameters in an operational setting.

Today, as information content increases and computer power grows to meet demand, LUTs are being supplemented by other methods to solve the radiative transfer equation. Techniques are being used to simultaneously retrieve multiple aerosol, gases, and surface parameters. These more flexible algorithms provide an alternative framework for retrieval, but they are still physically-based in radiative transfer. As we march towards 2043, the LUT methods will be permanently jettisoned, because although radiative transfer becomes more sophisticated with better representation of aerosol optical properties including irregularly shaped particles with complex composition, the dimensionality of the problem would need an infinitely large LUT. Thus, the trend towards optimal estimation, despite the need for an initial guess and the risk of finding local minima. With parallel improvements (and greater sampling) of ground-based and airborne observations including in situ sampling more properties can be constrained. The in situ measurements are especially important to characterize irregularly shaped particles. Algorithms will be designed to ingest data from multiple collocated sensors, especially combinations of passive and active sensors (Liu et al., 2017), and combinations of space-borne and ground-based sensor observations (Lopatin et al.,2013; Li et al., 2019). We expect joint retrievals of aerosol and trace gases, aerosol and land properties, aerosol and ocean color properties, and aerosol/gas/surface retrievals. Eventually there will be sufficient information by 2043 for retrieved properties to include vertical profiles of aerosol extinction and particle number concentration (Schlosser et al. 2022), retrievals over clouds (Torres et al., 2012; Jethva et al., 2013) and over all land surface types including snow and ice (Shi et al., 2019; Mei et al., 2020; Zhang et al., 2023). Characterization will include particle spectral absorption (Xu et al., 2021) and make progress towards constraining particle composition (Kacenelenbogen, et al., 2022).

By 2043, although radiative transfer and scattering processes will be better understood, completely
physically-based algorithms will be rare. Development of Machine Learning (ML) retrieval algorithms
will accelerate, such that there will be algorithms that produce a full array of aerosol characteristics by
pumping satellite-measured quantities through ML-derived models. These models have no physical
basis and do not rely on the radiative transfer equation (Eq. 1). Instead, the models are created by
training on a formulation data set that matches satellite-measured radiances with measured aerosol
characteristics. For example, today we can train a ML model using satellite measurements collocated
with AERONET observations (Lary et al., 2009, Di Noia and Hasekampf, 2018, Kang et al., 2022). In
this way we can obtain aerosol characteristics such as AOD that match AERONET with great accuracy
and precision (Lary et al., 2009), but we lose all physical understanding of how that AOD is retrieved.
The challenge for exclusive ML methods like these cited will be the identification of accurate
measurements of aerosol characteristics, other than AOD, in sufficient quantity to train the ML models.
However, ML algorithms will not recognize a physical situation that falls outside of its training data set,
and they still need to accommodate measurement uncertainty.

There are other uses of ML in aerosol remote sensing besides directly obtaining aerosol parameters
from satellite observations. Within a physically-based algorithm there are many assumptions that can be
aided by ML methods. For example, running a radiative transfer model is computationally expensive.
A ML model derived from radiative transfer runs will be more accurate than current interpolations of
LUTs and save time in an operational setting (Gao et al., 2021, Ukkonen 2022). In other work, ML
methods are used to classify specific aerosol types, such as dust by training on other satellite data sets
(CALIPSO) (Lee et al., 2021). ML is used to bias-correct aerosol products derived using standard
methods (Lipponen et al. 2022), and it can help with interpolation, constraining non-retrievable aerosol
optical properties, cloud (Wei et al., 2020) or snow masking, and surface reflectance determination (Su
et al., 2020). In 2043 aerosol retrieval algorithms will continue to be physically-based, but when
examined carefully will contain many elements of indispensable ML within.

## 4 The rise of the cyborg: Blends of measurements and models

The biggest noticeable change over the next twenty years will be the gradual extinction of "satellite
aerosol products". The entire purpose of what we know now as a satellite data product will be to feed
major assimilation systems that produce a complete representation of the global aerosol system within
the context of the complex Earth system (Zhang et al., 2008, Benedetti et al., 2009, Gelaro et al., 2017).
We already see the widespread acceptance of assimilation products, as noted by the number of citations

earned by key papers describing each system. For example, using Clarivate's Web of Science data base (Clarivate, referenced January 2023)) and searching on "satellite AND aerosol" we find the first returned reference to be Holben et al. (1998), the seminal AERONET paper with 5265 citations

accumulated over 24 years. Next up is Gelaro et al. (2017), the fundamental reference for the Modern-Era Retrospective Analysis for Research and Applications, Version 2 (MERRA-2) assimilation products, with 3270 citations accumulated over only nine years. We understand that not all citations to MERRA2 will involve applications to MERRA2 *aerosol* products and that having a single fundamental reference concentrates users' citations to a single paper, but the rate of acceptance of MERRA2 and

other systems' assimilation products over the past five years is astounding. See Figure 1.

Assimilation systems are analogous to the science fiction concept of the "cyborg", defined by the Oxford Dictionary of Science Fiction as "A hybrid being: half human, half machine (a contraction of 'cybernetic organism')" (Prucher, 2006). In our context, the assimilation systems are a hybrid of half

observations, half models. Satellites measure electromagnetic radiation that is processed through aerosol retrieval algorithms to determine aerosol parameters. Assimilation systems intercept parameters from the retrieval algorithms during processing to ingest that data into a global model that adjusts model representation of aerosol parameters to agree with the observations. Some assimilation systems draw from the aerosol retrieval algorithms close to the satellite observations, making use of radiance or

reflectance directly (Randles et al., 2017). Some systems ingest the final retrieval products such as AOD (Zhang et al., 2008). In all cases the global model is drawn closer to actual measurements at regular time steps and is expected to represent the actual aerosol system better than a model that is initialized and run forward from aerosol source inputs. The advantage of assimilation aerosol products is that users can access aerosol parameters for any time period, at any location, at any altitude. The parameters will

have the benefit of ALL available data from ALL satellites simultaneously. User's will not need to patch together products from different algorithms. The model does that for you and presents the results in an Earth system governed by physical restraints.

The disadvantages of assimilation are that the resulting aerosol field can be variable, changing abruptly

with model time step or with the available of different aerosol input fields. For example, there will be much more data available during the day than at night, which may create day/night discontinuities. Furthermore, by drawing the model nearer to the observed variables there can be arbitrary introduction or subtraction of aerosol mass. This can create biases in mass budget parameters, such as mass deposition. Furthermore, model parameterizations such as chemical transformation and interaction with

clouds remain uncertain, and likely will remain uncertain in 2043. Nevertheless, assimilation data sets

will become the norm for a majority of aerosol data product consumers. The assimilation systems will run in the cloud and will be called on to produce the cyborg data at user's will.

We note that satellite-retrieved aerosol products can interact with models without being assimilated. They can serve as validation and identify and characterize aerosol point sources such as smoke and volcanic aerosols. Using satellite-derived data sets as separate, independent data can improve parameterization of aerosol processes within models. Such processes include chemical transformation and aerosol-cloud interactions. Such independent constraints on models have important implications for climate prediction and air quality applications (Kahn et al., 2022).


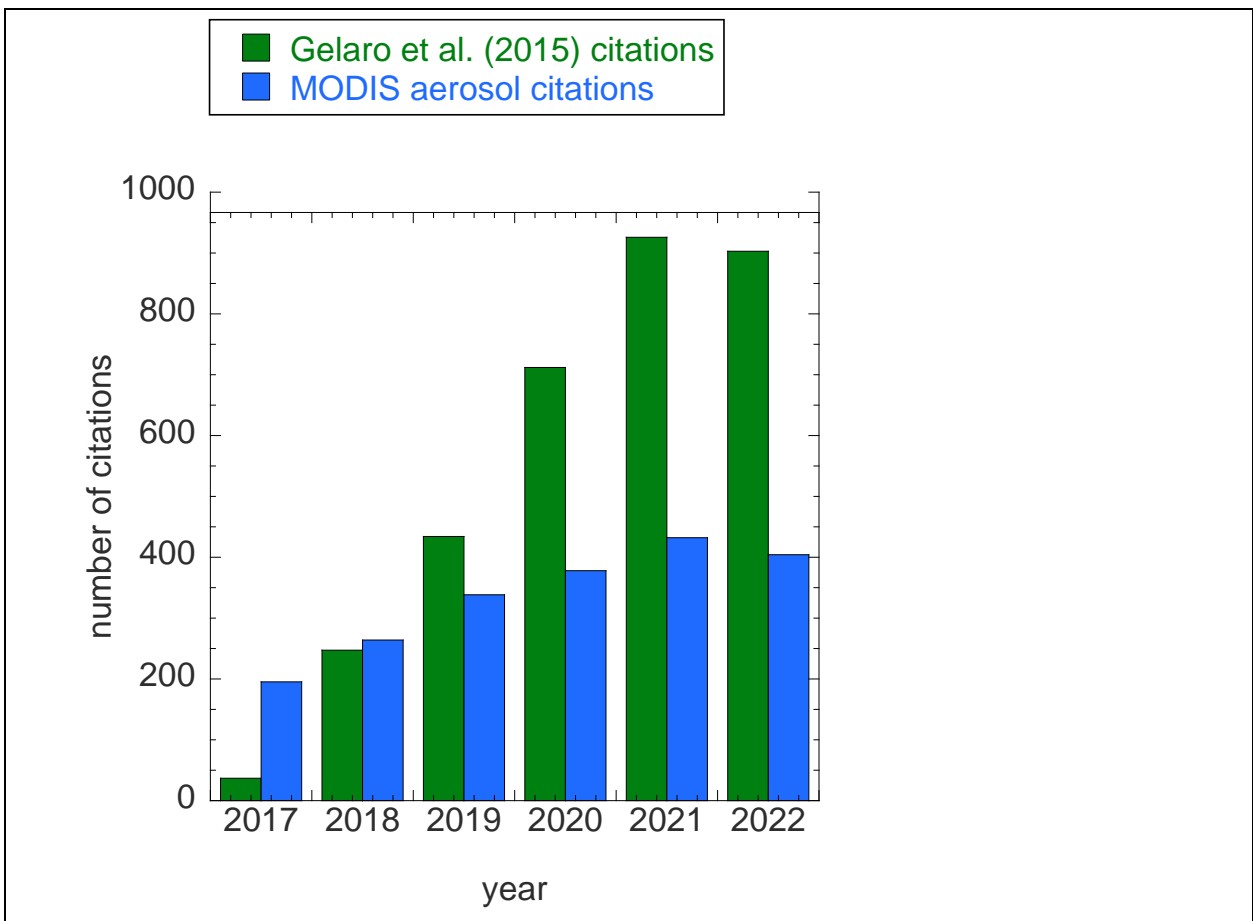

Figure 1. Number of citations received by Gelaro et al. (2017), the fundamental reference for the NASA MERRA2 assimilation system products (green) and the combination of citations received by Levy et al. (2013), Hsu et al. (2013) and Lyapustin et al. (2018), representing the Dark Target, Deep Blue and MAIAC aerosol algorithm products for MODIS (blue). Source: Clarviate Web of Science data base. The point is to note the rapid

acceleration of MERRA2 citations as an indicator of the speed of the research community's acceptance of assimilation products, while traditional retrieval products such as MODIS aerosol products maintain a consistent presence in the research community. We anticipate that over the next twenty years, assimilation product use will continue to grow while use of standard products will plateau and decrease.

## 5 Satellites, satellites everywhere

A gold rush to space has begun and will intensify over the next 20 years as space becomes crowded with a variety of sensors in various Earth orbits. During this period we will realize a true democracy in space as non-traditional players such as the space agencies from the developing world and from a wide range of commercial sector companies join traditional national agencies in blanketing the skies with satellites and sensors. While only a fraction of these satellite sensors will be Earth-viewing and aerosol-

capable, the result will yield a tremendous volume of data useful for aerosol retrieval.

Satellites will come in all sizes. Traditional agencies will still launch large, expensive, robust flagship missions with multiple platforms and multiple sensors on each platform. The flagship missions will continue demonstrating lower risk, higher accuracy, greater reliability and longevity. The flagship

missions will be supplemented by and integrated with constellations of smaller satellites and nano-sized cubesats. The smaller satellites will be orders of magnitude less expensive, and thus riskier and with more limited capability than the larger observatories. However, the multitude of the smaller units renders the loss of any particular unit as insignificant to the greater objectives. With so many satellites flying, observing Earth and returning measurements for aerosol retrieval, each satellite will be seeking a

particular niche. All orbit types will be covered including polar-orbiting, inclined orbits and geosynchronous, and near real time observations will be achieved.

The challenge of so many data sources is the maintenance of data quality. There will be a lot of data, and there will be a lot of bad data. Eventually the scientific and user communities will organize to set

standards and protocols to regulate data quality. Such standards will rely on the cohesion of the data communities for enforcement. An example of a community establishing standards that are enforced through peer review are the protocols published by the International Ocean Colour Coordinating Group (IOCCG) for satellite ocean colour sensor validation. Not all data will be free and open access. Commercialization of data products will aid in enforcement of standards, as different entities will be

competing to sell their products based on many characteristics including quality. Evaluation of that quality may move into the hands of independent companies or coordinating groups. Commercial entities

that launch satellites intend to produce specialized data products for specific customers. Sometimes these customers will be national agencies looking for input to their assimilation systems. Other times commercial entities will be producing their own assimilation system cyborg products and selling those results to their customers.

## 6 Calibration and Validation systems will need to keep up

Calibration will be a continuing concern with the proliferation of small and nano satellite data flooding the market. As stated above, there will be a lot of bad data, much of it due to poor calibration of each individual sensor, as evidenced by today's fleet of mostly poorly calibrated small satellite sensors. This will change. Calibration takes many forms and includes radiometric, polarimetric, geometric, spectral components and each component requires its own intense scrutiny that not all satellite-launching entities are ready to provide. To mitigate some of the chaos, a commercial pre- and post-launch calibration industry will spring up. These companies will offer pre-launch calibration facilities and take responsibility for a sensor's calibration, even verifying that calibration using vicarious methods once the sensor is on orbit. Eventually, either governmental or space industry bodies will move to a certification process similar to the Underwriters Laboratories (UL) or Conformite Europeenne (CE) marks common today to denote adherence to a specified set of standards for electronic systems. There will also be a need for in-space calibration options, such as was proposed for the Climate Absolute Radiance and Refractivity Observatory (CLARREO) mission. CLARREO would maintain highly accurate, traceable reflectance measurements from the Earth and Moon through its lifetime, which could be transferred to other instruments in orbit.

Validation of aerosol products is another issue. With the democratization of space, the leading national space agencies will no longer hold a monopoly on space access. Ironically, instead these government agencies that pioneered aerosol characterization from space will take the lead on *suborbital* measurements to support the validation side of producing high quality aerosol products. Government-sponsored field experiments and ground networks will provide the basis for the entire international community, including the commercial sector, to evaluate their products and improve their algorithms

Today's basis for validation of aerosol data sets is the ground-based AERONET network (Holben et al., 1998, Giles et al. 2019). The AERONET concept will continue for decades into the future, but the technology itself will evolve. Instruments will move from finite wavelength bands to hyperspectral spectrometers and will continue AERONET's current trend into the shortwave infrared. Originally, AERONET instruments had invested in polarimetry, which turned out to be a capability ahead of its

time. The community has caught up and AERONET will expand the polarimetric capabilities of their networks and incorporate polarimetry into their retrievals. Data capacity, storage and transmission will expand, as will the network itself. AERONET's subsidiary, the Marine Aerosol Network (MAN) will continue to collect data on ocean-going cruises and ships of opportunity.  The MAN instruments will also upgrade their technology and data capabilities. The AERONET retrieval that produces reliable aerosol characterization in addition to AOD, including particle size distribution, single scattering albedo, complex refractive indices, and non-sphericity, will become a more important asset over time as aerosol satellite remote sensing advances in information content, requiring evaluation of satellite-retrieved products that include a wide range of aerosol parameters. However, we note that while the uncertainty in AERONET AOD is sufficiently small to offer true validation to a collocated satellite AOD product, the uncertainty of AERONET inversion products (particle size distribution, single scattering albedo etc.) is not.  The AERONET inversion products are themselves subject to assumptions and caveats, and may have greater error than certain satellite parameters. Still, the inversion products provide a vital service in the evaluation of satellite products, providing a standardized framework of high quality, widely distributed aerosol characterization for nearly immediate comparisons with satellite products.

The AERONET concept of a widespread, high quality, aerosol observing ground-network will proliferate, as we are already witnessing with the Interagency Monitoring of Protected Visual Environments (IMPROVE:  Malm and Hand, 2007), the European Aerosol Research Lidar Network (EARLINET: Pappalardo et al., 2014), the Aerosol, Cloud and Trace gases Research Initiative (ACTRIS, 2023), the Surface Particulate Matter Network (SPARTAN: Snider et al., 2015) and other networks. The different networks offer different types of aerosol characterization necessary to keep up with the expansion of retrieved aerosol parameters from space. In-situ measurements of aerosol aerodynamic properties will continue to be essential for the assessment of the aerosol health impact and to link aerodynamic to optical measurements, which form the basis of satellite products.  Ground LIDAR networks (Welton et al., 2001; Pappalardo et al., 2014) assisted by improved airport ceiliometers will expand high temporal characterization of the aerosol vertical profile.   As satellite retrieval algorithms upgrade their radiative transfer, refine their assumptions, incorporate ML and become cyborgs, so will the ground network retrieval algorithms.  More and more often validation data sets will include the resulting aerosol characterization obtained by simultaneous retrieval of several collocated instruments.  Ideally this will include passive measurements with collocated lidar and ground-level or balloon-borne in situ measurements.

Field experiments featuring aircraft-borne measurements will be indispensable for validating the
satellite observations at the radiance/polarization level (Level 1) and the retrieved aerosol particle
properties in regions and situations of high scientific interest or societal relevance. Aircraft fly in three
dimensions, sampling situations between network stations and providing vertical profiles of parameters
where lidar is absent. Field experiments will also be essential for confirming ground-based validation
sites, as now these sites will be producing a suite of retrieved parameters that require their own
validation.  We will find that field experiment measurements will be the only truly independent data set
for validation. With the abundance of suborbital data, quality again becomes an issue, and national
agencies will be the only community players with the resources to maintain high quality, long-term
validation data sets.

At the opposite end of the spectrum will be the proliferation of low-cost citizen science networks,
including use of smart phone imagery. These commercial networks will expand across populated areas
and will contribute to characterizing the global aerosol system simply by the abundance of available
observations.  However, there will not be the same oversight to maintain quality of these data sets and
users will need to calibrate the data from citizen scientist measurements with government-maintained
and quality-assured sources.

## 7 Unanswered science questions will drive the new systems

We have described a vibrant global measurement system focused on aerosol remote sensing as we move
forward into the next twenty years. This picture assumes a continuing need for aerosol characterization,
which will be driven by pressing science questions and societal priorities.  What will be the aerosol
science questions of the coming decades?

Quantifying changes to the aerosol system will be essential as we continue to struggle to catch up to
climate change. By 2043 we will have 70 years of Total Ozone Measuring System (TOMS), 60 years of
AVHRR and 40 years of MODIS + VIIRS aerosol time series. While most of these time series offer
insight into global and regional changes to aerosol loading, inferring changes to aerosol types and
particle properties will be especially important.

We will still be working towards understanding aerosol-cloud-precipitation processes, how these have
changed in time and how do these processes interact with atmospheric dynamics.

The role of aerosols in transporting nutrients around the globe and depositing them into terrestrial and oceanic ecosystems will continue to be a pressing issue in the coming decades. Likewise, the role of aerosols in transporting pathogens will still be under investigations in the coming years.

Maintaining healthy air quality for all the word's populations will continue to call for understanding of global aerosol distributions and characterization in a way that will require satellite remote sensing.

Underlying these issues will be the goal of using the characterization of the global aerosol system to make our planet healthier and environmentally just.

**8 So what will 2043 look like?**

Despite today's assessment that we are "data rich", by 2043 we will look back at 2023 as a primitive data desert. Aerosol products that are actually cyborgs produced by assimilation systems will represent the global aerosol, producing near real time representation of aerosol fields at fine temporal and spatial scales. Assimilation aerosol product users will forget that clouds (real clouds not data clouds) exist, as
all aerosol fields will be interpolated through traditionally cloudy scenes by the models. Likewise, assimilation aerosol data fields will have no gaps due to difficult surfaces such as sun glint, snow or ice. However, the challenge will be the quality checking and synthesis of so much data. There will be commercial aerosol data ambassadors who, for a fee, will create custom data packages from the overwhelming supply of public and private data product sources.

Today's retrieval algorithms will appear quaint to aerosol remote sensing scientists in 2043. The days of a single inversion applied to a single sensor will be in decline. Even physically-based algorithms will be designed to invert the observations from multiple sensors and platforms simultaneously. This includes joint inversions between satellite and ground-based sensors to produce full characterization of
atmospheric constituents (aerosols and gases), as well as complete characterization of the surface beneath whether that surface be a cloud, the ocean or a snow field. The caveat with multi-sensor inversions is the difficulty of determining the propagation of error and uncertainty from sensors of various capability and calibration.

Few people will care about aerosol optical depth. The assimilation systems will offer aerosol characterizations in forms most useful for individual users. Air quality managers will automatically obtain particulate mass concentration. Aerosol-cloud-precipitation scientists will obtain vertically-resolved number concentration or heating rates due to aerosol absorption. Those interested in aerosol

radiative forcing, will find numbers in $Wm^{-2}$ coming automatically out of some assimilation system.

There will be attempts to produce deposition mass flux, but the error bars will still be large. Likewise, attempts to reliably retrieve chemical composition directly from remote sensing likely will fail, although aerosol type will be constrained very well.  The only interest in aerosol optical depth will be users working towards continuity with the old sensors for long-term trend analysis or as a critical first step in validating a retrieval.


In 2043 validation networks must be in place and major research efforts will switch from algorithm development to designing and implementing a strategy that reserves sufficient independent data for validation. If the assimilation is producing a cyborg vertically-resolved distribution of particle number concentration near or in clouds, then there needs to be a validation strategy to confirm and put error bars

on that product. Investment will be needed to develop the instruments and platforms needed to implement a validation strategy. True validation of some difficult parameters such as vertically-resolved particle number validation may be measured only infrequently during dedicated field campaigns. In some ways validation in 2043 may look like a step backward from what aerosol scientists have come to expect today.  Instead of 1 million collocations that we achieve now when we validate AOD with

AERONET, for some retrieved aerosol parameters, we will be satisfied with 10 to 50 validation points hard-earned in the field.

Where will the funding come from? From the same tax base that has propelled major space agency missions in the past, but now directed more to assimilation systems, validation, quality assurance and

buying data from commercial entities who can sell it cheaper than the agencies can produce themselves. In addition, there will be private concerns, investors and customers, who see value in data products tailored to particular concerns: the insurance industry, the agricultural industry, local air quality jurisdictions.

In summary, 2043 will be an exciting time for aerosol remote sensing.  There will be plenty of data for answering science questions and plenty of work to do to assure the integrity of that data.  We are sorry to say, though, that we cannot promise flying hover boards as seen in the *Back To The Future* movies (Zemeckis and Gale, 1989, Schildhouse, 2014), even in 2043.


**Author contribution**: The manuscript resulted from discussions between the three authors, over a time span of years. LR wrote the initial draft.  RL and JVM edited this draft and added their own insights. LR

took responsibility for addressing reviewers' comments with RL and JVM editing her responses and subsequent versions of the Opinion.

**Competing interests**: LR and JVM hold ownership stakes in GRASP SAS in addition to their primary employment at the University of Maryland Baltimore County. GRASP SAS is involved in the commercial small satellite industry and in producing aerosol remote sensing products from its own satellites and from publicly available satellite measurements. GRASP SAS's subsidiary, Airphoton Inc., designs, builds and sells ground-based instruments for aerosol measurements and payloads for space.

**Acknowledgements:** The authors thank both reviewers, Dr. Ralph Kahn and Prof. Zhanqing Li for their very thoughtful comments that caused us to push harder towards a defensible Opinion. Many of the opinions of these reviewers were incorporated into our Opinion in its final form, and we are proud to acknowledge them both by name.

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
