# Peer review of "Opinion: Aerosol Remote Sensing Over The Next Twenty Years"

_EGUsphere, 2023_

## Author Response (AR1)

This is a response to Reviewer 1's comments.  The reviewer's comments are in black font.  Our responses are in blue font.

This is a clearly labeled opinion piece.  The role of a reviewer on such a paper is different from that when reviewing a research or review article. The authors are of course entitled to their opinions; the reviewer need not agree with them, and a reviewer's opinions are not necessarily any more legitimate than those of the authors.  So, I have included a few specific factual notes, but my approach in this case is primarily to suggest thoughts that occur from my perspective, for the authors to consider if they wish.

We thank Reviewer 1 for their time in reading our speculative Opinion. Our goal was and is to generate discussion within the community, and the Reviewer's response is a great start to this discussion!

- Around line 124, ff. Won't there be tradeoffs, even in the (foreseeable) future, among data rate, spatial resolution (which is really critical for many applications), spatial coverage, and the number of spectral channels? I'm wondering whether "hyperspectral" would always be the best choice if it comes at the expense of spatial resolution or coverage. I know there are some ideas about on-board data compression, but there are limits, and also some disadvantages. An estimate of the required data rates compared to current and anticipated downlink and compression capabilities might be helpful at bringing some reality to the tradeoffs here.

We thank the Reviewer for jumping right in! This is exactly the reaction we were aiming for.

There are always tradeoffs.  But the picture we are trying to paint is that with so many new players in the game the choices will expand. It won't be necessary to design, build, launch (and egads, pay for) the perfect aerosol mission where each trade and descope is painful, knowing that this perfect mission is our last chance for the next 10 to 20 years. Instead, you have a fleet of nano satellites measuring multispectral radiometry at very high resolution. You have another constellation that focuses on multi-angle polarimetry, and another set of hyperspectral measurements. Then along comes lidar of various types, and all the data is ingested into assimilation systems to provide seamless 4-dimensional depictions of the global aerosol system. And every two months there is another Space-X launch putting another 40 small satellites into space. The pressure to be perfect dissipates, and trade offs are less vexing.

We do address the Reviewer's specific issues about data rates and other issues further down in the discussion.

- Line 137. Perhaps a future multi-angle polarimeter instrument will improve signal/noise for airborne dust plumes, but you might consider the substantial capabilities of the current EMIT mission for the dust mineralogy application. As this is an opinion piece, do you have an opinion on what the incremental value of another dust mineralogy mission would be compared to other future mission options?

  We didn't even think of EMIT because we are so focused on the retrieval of airborne aerosol particles. We make that clear at the end of the paragraph of the line cited by adding the word "airborne". We also mention the omission of EMIT before introducing the new Table 1.

- Line 154. Do you have a sense for the likelihood of overcoming the technological issues involved in deploying the "ideal" lidar within 20 years? If so, would the cost be within the range of reasonableness?

We remain optimistic about the $3\beta + 2\alpha + 1\delta$ lidar. As we see from the new Table 1 the next two lidars will be HSRL, one from Europe and one from China. The Chinese one, ACDL, is already flying, although we haven't seen publications yet making use of that data. China is planning two ACDLs, running until 2033. ACDL is definitely not $3\beta + 2\alpha + 1\delta$, but 2043 is another decade beyond the end of the ACDL era. This is sufficient time for the next generation of aerosol lidar to be designed and launched. Yes, it will be expensive and we note in the text now that NASA's AOS HSRL was descoped because of cost. NASA does things very expensively, but we would not rule out the commercial sector in producing the first $3\beta + 2\alpha + 1\delta$ lidar with higher risk but at much reduced cost. However, the tone of the lidar paragraph may be too enthusiastic and so we change the "will"s in the lidar paragraph to "could"s to soften the tone.

But we are very happy to have the Reviewer engaged!

- Lines 182-187. I think you need to estimate the actual data rates involved, and provide some indication as to how these compare with a reasonable expectation of what might really be possible within 20 years. You mention many new technologies, but do not discuss the limits on their expected capabilities. To be deployed by 2043, these technologies would have to be ready for space-qualification no later than 2034-2036, which is not so far away. I appreciate that you are laying out "everything imaginable," but I think the presentation would be much more useful if that were also contrasted with "everything achievable" as well as "everything likely." (I note that the primary new NASA mission in this area for at least the next 10 years was recently descoped severely.)

We see technology advancing quickly. A multi-angle polarimeter produces 15 to 60 times the data per pixel per wavelength than does MODIS.

3 polarization states X 5 along track view angles as per PACE=SPEXone as an example minimum number = 15X

3 polarization states X 20 along track view angles as per PACE-HARP2 for a max number = 60X

Even factoring in the number of wavelengths, the data rates are still within the 15X to 60X range of MODIS.

The Ka band on the PACE observatory will be able to keep up with these data rates. However, 20 years from now, there will be even more data. NASA released two announcements earlier this year.  One was soliciting for people interested in licensing new technology for Ka band communications.  The other was a news release announcing success with an optical communications system. The difference between optical and radio technology in data downlink rates is three orders of magnitude. Optical systems will be the future for satellite-to-ground communication as data volumes increase.

As we stated, our intention is to start a discussion.  While we are skewing to the optimistic, everything we have presented falls into the "everything achievable" category. We see the technology demos now that are proving the concepts that we discuss, and we see first efforts of these technologies already flying in space. To go from "everything achievable" to "everything likely" takes will and determination to get there. We will remain optimistic that the determination exists.

We explicitly mention the two NASA announcements in the text.

- Lines 305-312. It might be worth mentioning something about the future of radiometric and possibly also geometric calibration for all these instruments. The radiometric accuracy required to perform aerosol optical depth retrievals is challenging for many small-sat instruments, and the calibration requirements for meaningful aerosol-type retrievals are considerably more stringent.

  We did say that "there would be a lot of bad data". However, we believe that it is possible for nano-sized satellites to improve to the point that aerosol retrievals will be possible. As the Reviewer states, calibration will be a major factor. We have decided to explicitly speculate on what calibration effort will

look like when space is filled with small satellites. We have changed Section 6 to: **Calibration and Validation systems will need to keep up** and inserted a full paragraph on calibration ideas.

- Lines 316-317. This is an intriguing idea that might warrant elaboration. So, you are suggesting that commercial pressure will "enforce" standards? We have certainly seen a lot of satellite data assessment over the past two decades, but nothing remotely like enforcement of standards, or even agreed-upon standards for reporting uncertainties. And in many cases, there is a political component to the way data quality is reported. (In a commercial context, it would be called "false advertising" – in practical terms, what entity could counter that in the satellite aerosol data field?)

  We have two examples.  The first is self-regulating community enforcement as demonstrated by the ocean biology community. There is the International Ocean Color Coordinating Group (IOCCG) that has created a consensual set of protocols for ocean color sensor validation.  https://ioccg.org/what-we-do/ioccg-publications/ocean-optics-protocols-satellite-ocean-colour-sensor-validation/. It has become difficult for researchers to ignore these protocols during peer review, either for papers or proposals. Enforcement is peer pressure.

  The second example is consumer enforcement. We mentioned UL and CE standards for electronic systems in the new calibration paragraph.  The UL certification is an industry standard, while the CE certification is adherence to a government standard. Obtaining these certifications costs a business money, but customers look for that certification in purchasing.  In Europe to not have CE certification means to forego selling products to any government-sponsored customer including electronics purchased by scientific grants.  In this way governments can enforce standards.  We see a combination of these types of enforcements starting to materialize. Can't you visualize ECMWF saying, "We can't buy your data and assimilate your product unless the XX certifies the accuracy" ?

  We have added mention of the IOCCG in the text.

- Line 390. Aerosol-cloud interaction is a comparable or possibly larger factor affecting climate than aerosol direct and semi-direct effects, yet we do not even have measurement approaches to adequately constrain many of the

underlying mechanisms involved at present. Any thoughts on the future of such measurements – for water and especially for ice clouds?

Oh, we agree, and if we thought a bit we could likely come up a whole other Opinion on the technology and infrastructure needed.  Come find us at a conference and we can speculate more together.

But one topic at a time, and this one is aerosol remote sensing.

Lines 406-408. It seems reasonable to anticipate *some* unexpected advances will occur in the next 20 years, but to assert that all these challenges will somehow be met seems unreasonable. Thinking back 20 years, we have come a long way, but many of the expectations and promises from that time have not been met either. Any idea at all how we actually get from here to there? Addressing this, and perhaps qualifying the predictions accordingly, as needed, would greatly strengthen this paper in my opinion.

The point here is that these traditional challenges will be overcome through assimilation systems and the cyborgs, not that aerosol remote sensing itself will start retrieving through clouds, over glint etc.  How do we get from We stop beating our heads against walls to produce pure aerosol retrievals in too challenging situations and instead work on the best ways to keep assimilation products honest.

Even though this paragraph is about assimilation products, we do note that there are studies making progress in expanding retrievals from satellite. Some of these references are given in the 3$^{rd}$ paragraph of Section 3.

We think there was some confusion about our intention in this paragraph and have made some wording adjustments.

- Lines 425, ff. Why such a modest start at considering the limitations, and only at the very end? I'd add that for the entire history of the Earth-observation from space, and much longer and in many more places than that, cost has been a limiting factor, if not *the* limiting factor. Have you made any rough estimate of how the cost of everything conceived here compares with current budgets for such work, or what funding sources could possibly meet the estimated requirements?

Cost is always a factor. But… we are seeing small steps towards the picture of 2043 that we describe here, including the ambitious Chinese program for HRSL and the European commitment to multi angle polarimetry through 2046. The other point is that we are seeing an explosion of interest from non-traditional players in the game, whether they be emerging national space agencies or the commercial sector. These non-traditional players do not have the budgets that NASA and ESA have used to launch costly missions of the past, and yet the non-traditional players are pushing forward with missions and products.

Two of the authors of this Opinion, Remer and Martins, have worked on big national agency missions and are also involved with a private company in which they have financial interest. The private company builds small space payloads and derives aerosol products from sensors launched on nano satellites. Are the company's aerosol products as good as the ones from national agencies?  Not yet. But they are produced at less than 5% of the cost of traditional products.  Given twenty years, the expectation is that quality private sector products will increase to meet users' needs.

Where does the funding come from? From the same tax base that has propelled major space agency missions in the past, but now directed more to assimilation systems, validation, quality assurance and buying data from commercial entities who can sell it cheaper than the agencies can produce themselves. But the funding also comes from customers who find value in customized data for their own purposes: the insurance industry, the agricultural industry, national and local environmental agencies and jurisdictions.

The Reviewer is correct in that cost is a factor, but we have seen so much cost reduction from thinking small and accepting risk that we do not see cost as the same impenetrable barrier as does the Reviewer. We are very happy to have his opinions.

We have added a short penultimate paragraph that discusses future funding.

**More Specific Notes**

Line 46. You might consider adding the reference: Hammer et al., 2020. Global estimates and long-term trends of fine particulate matter concentrations (1998-2018). Environ. Sci. Tech. 54, 7879–7890, doi:10.1021/acs.est.0c01764.

Done.  Thanks.

Line 87.  Might be: "… if the aerosol products are accessible, validated, adequately documented, and part of a larger…" Documentation so a user can actually understand the strengths and limitations of a dataset is non-trivial.

Absolutely non-trival. Clause added.

Line 120. I think it is fair to say MAIA is expected to launch in the mid-2020s (last I heard was some time in 2024).

Yes.  Still listed as 2024.  Changed to "mid"

Lines 125-127. Note that the range of observable scattering angles diminishes away from the sub-spacecraft point for a broad-swath imager, so for broad, multi-angle coverage, many imagers would be needed.

Fewer scattering angles, but still more than a single view MODIS. We believe our wording is acceptable and have not made any changes.

Line 146.  Is an HSRL still being considered for AOS?

No. And we now explain that it was descoped for budget concerns, but the Chinese have one flying now.

Line 204. Might be: "… an acceptable error, consistent with measurement uncertainty." Just to emphasize that the entire process is also limited by measurement uncertainty (which all too often is not well characterized).

Good point.  Thank you.

Lines 209-211. This is true only if the additional wavelengths contain some orthogonal constraint on the surface or atmosphere. For multi-spectral data in general, there is often a lot of redundancy for this application. Wavelength selection and sensitivity become key considerations when instruments are designed.

Very good point.  Added explanation and a reference.

Lines 221-222. Cost-function minimization can be done in a LUT framework too. Also, optimal estimation approaches are certainly important tools, but it is not clear that they will require fewer assumptions. For example, there can be many local minima, and

especially when the derivatives are small, and the solution can depend heavily on an initial guess.

Thank you. Changed the wording there.

Lines 224-225. Characterizing irregularly shaped particles optically, and many other particle properties, will require *in situ* measurements; there are physical limits to what can be retrieved from even ideal aerosol remote-sensing observations. (It is of course different for gases.)

We agree and have added a sentence mentioning research using in situ measurements.

Lines 244-245. Perhaps more importantly, ML models cannot retrieve (or in some cases, even recognize) when the physical situation is outside the parameter range of the training set. And again, the measurements will have uncertainties, and there are also physical limits to the particle property information that remote-sensing measurements can provide.

Sentence added at the end of the paragraph.

Lines 261-262. I think this is an overstatement. Assimilation is one way models can ingest satellite measurements, provided the measurements are sufficiently well-sampled, and there is a formal uncertainty associated with each measurement. However, constraining models with other satellite measurements can require different approaches, such as applying observations that could be used to characterize volcanic or wildfire smoke plumes, or other discrete aerosol sources.

Short paragraph added at the end of the section.

Lines 286-291. Assimilation is an important technique, but to be fair, the paragraph here might point out that models have other limitations that are not addressed by assimilation. Much about the underlying parameterizations in models is also assumed, such as aerosol removal process efficiencies, the mechanisms for a range of aerosol-cloud interactions, many of the chemical and physical transformations that mediate particle aging, etc. Adequately constraining these for climate or air quality applications cannot be done with remote sensing measurements alone.

A paragraph on limitations of assimilation has been added.

Line 355. You might consider adding the reference: Welton, et al., Proc. SPIE 4153, Lidar Remote Sensing for Industry and Environment Monitoring, (13 February 2001); doi: 10.1117/12.417040

Thank you.  Reference added.
* * *
Response to Prof. Zhanqing Li.

We present below the Reviewer's comments in black and our response in blue.

The invited paper was well written by three distinguished experts in aerosol remote sensing, thanks to their rich knowledge and experiences.  Besides presenting an overview of major milestones in the subject, the authors provide their vision in the development of aerosol remote sensing in the next 20 years that are very valuable.  As an opinion paper, it is acceptable to express their personal views on the evolution of aerosol observation technology, platform, products and their applications for a total period of four decades. Having realized this, I'd not apply the usual scientific rigor to gauge the paper's quality and suitability for publication in the ACP.  On the other hand, however, we ought to make sure that any history be portrayed correctly and properly, and the future be projected with fidelity and feasibility, not in violation of any fundamentals of physical principles.

The authors thank Prof. Li for his comments and appreciate his efforts to assure the accuracy of the history we present and temper our speculation of the future.

Our goal in writing this Opinion was to provide the opening move in what we hope will be a community-wide discussion on the future direction of aerosol remote sensing. The decadal surveys provide an essential measured and feasible roadmap for the future. We did not intend to duplicate that effort. Instead, we decided to offer the results of speculative brainstorming to provide the seed for discussion.  Yes, the result is often too optimistic, but in our opinion a bit of optimistic provocation is good for the community. People need to dream.

Still, we respect Prof. Li's sentiments here, and we have adjusted the text.

**Major comments:**

Some statements are overly too optimistic without giving any basis such as:

2043 or in 20 years from now is supposed to be the main theme of "opinion". While it is personal opinion, a review of planned activities especially those by governmental agencies of space-leading countries (US, EU, China, etc) would make such opinions more trustworthy. While 20-year is a long period, it's just two cycles of decadal survey during which a lot have been planned and thus many are already in different stages of development. A comprehensive review of such "officially" planned aerosol remote sensing activities should be added.

We had done a cursory discussion of some planned missions in Section 2 but agree that there is room for more. We have added Table 1 to Section 2 and provide additional text in that section. However, there is no way to be comprehensive. The remote sensing community is expanding too quickly for that.

L229: "sufficient information by 2043 for retrieving the vertical profiles of aerosol extinction and particle number concentration, retrievals over clouds and over all land types including snow and ice". None of these are trivial to do, and most are so far-fetched that I'd doubt it is feasible in 20 years unless if there is any black technology in the horizon. In my view, profiling of aerosol extinction is mostly within our reach in this time frame, but it'd be impossible to get aerosol size distribution on the global scale, let alone at different levels. Besides active remote sensing, it will remain as a seriously ill-posed problem over bright surfaces, except for some being spectrally bright/black surfaces.

We are excited that Reviewer Li is responding to our speculation with his own opinion!

Prof. Li agrees that vertical profiles of extinction are not too farfetched.

What about vertical profiles of particle number concentration? When we were involved with the formulation study group for the potential NASA Aerosol Clouds Ecosystems (ACE) mission in the 2000s, there was discussion about the need for vertical resolution of particle number concentration. Sensitivity studies performed then suggested it could be possible with sufficient aerosol signal using HSRL. Furthermore, a combined retrieval using HSRL with a multiangle polarimeter yielded even more robust vertical characterization. A later example of this theoretical work was compiled in a NASA

Technical Memo by Liu et al. (2017). More recently, Xu et al. (2021) have demonstrated with airborne HSRL and polarimeter vertical profiles of single scattering albedo (SSA) and size distribution. Schlosser et al., (2022) use the same airborne instruments in a different field campaign to derive vertically-resolved particle number concentration. To us, it seems optimistically possible that the success of the airborne retrievals will be translated to space within the next 20 years, especially since the Chinese reportedly have launched the first HSRL in space (ACDL aboard DQ-1).

Retrievals over clouds are now common aerosol products using multispectral radiometers with spectral range in the ultraviolet (Torres et al., 2012) and even the visible (Jethva et al., 2013). Add in lidars, polarimeters, hyperspectral with oxygen-band retrievals, and we see great potential for continued robust aerosol characterization over clouds.

As for difficult surface types, we agree that our statement is a bit of a stretch. However, when we started with MODIS, we avoided deserts. Now deserts pose little difficulty. Snow and ice are still trouble, but people have begun to try ideas: Shi et al. (2019), Mei et al.(2020), Zhang et al. (in review 2023).

How much of this will be global by 2043? We doubt that much of it will be truly global. Advanced retrievals will require substantial aerosol signal. We are not suggesting vertical profiles of particle number concentration for AOD = 0.05, but as signal increases and the number of "eyes" on each scene increases, retrievals that combine multiple sensors will begin to produce parameters that in the past seemed impossible. We stand by our opinion.

But we have modified the text with additional references to better support that opinion.

Shi, Z.; Xing, T.; Guang, J.; Xue, Y.; Che, Y. Aerosol Optical Depth over the Arctic Snow-Covered Regions Derived from Dual-Viewing Satellite Observations. *Remote Sens.* **2019**, *11*, 891. https://doi.org/10.3390/rs11080891

 Zhang, Z., Fu, G., and Hasekamp, O.: Aerosol retrieval over snow using RemoTAP, Atmos. Meas. Tech. Discuss. [preprint], https://doi.org/10.5194/amt-2023-127, in review, 2023.

Xu, F., Gao, L., Redemann, J., Flynn, C.J., Espinosa, W.R., da Silva, A.M., Stamnes, S., Burton, S.P., Liu, X., Ferrare, R., Cairns, B., Dubovik, O.: A combined Lidar-Polarimeter inversion approach for aerosol remote sensing over ocean, Frontiers in Remote Sensing., 2, https://doi.org/10.3389/frsen.2021.620871, 2021.

Schlosser, J. S., Stamnes, S., Burton, S. P., Cairns, B., Crosbie, E., Van Diedenhoven, B., Diskin, G., Dmitrovic, S., Ferrare, R., Hair, J. W., Hostetler, C. A., Hu, Y., Liu, X., Moore, R. H., Shingler, T., Shook, M. A., Thornhill, K. L., Winstead, E., Ziemba, L., and Sorooshian, A.: Polarimeter + Lidar − Derived Aerosol Particle Number Concentration, Frontiers in Remote Sensing, 3, 885332, https://doi.org/10.3389/frsen.2022.885332, 2022.

Mei, L., Vandenbussche, S., Rozanov, V., Proestakis, E., Amiridis, V., Callewaert, S., Vountas, M. and Burrows, J.P.: On the retrieval of aerosol optical depth over cryosphere using passive remote sensing. *Remote Sens. Environ.*, **241**, 111731, https://doi.org/10.1016/j.rse.2020.111731, 2020.

Liu, X., Stamnes, S., Burton, S, Ferrare, R., Hostetler, C., Chemyakin, E., Mueller, D., Cairns, B.: A Combined Polarimeter and Lidar Optimal Estimation Algorithm to Improve Aerosol Microphysical Property Retrievals. NASA Technical Reports Server, Document ID 20200009791, https://ntrs.nasa.gov/citations/20200009791, 2017.

**Torres, O.**, **H. T. Jethva**, and **P. K. Bhartia**. 2012. "Retrieval of Aerosol Optical Depth above Clouds from OMI Observations: Sensitivity Analysis and Case Studies." *Journal of the Atmospheric Sciences* **69** (**3**): 1037-1053 [10.1175/JAS-D-11-0130.1]

**Jethva, H. T.**, **O. Torres**, **L. Remer**, and P. K. and Bhartia. 2013. "A color ratio method for simultaneous retrieval of aerosol and cloud optical thickness of above-cloud absorbing aerosols from passive sensors: Application to MODIS measurements." *IEEE Transactions on Geoscience and Remote Sensing* **Vol 51** - 3870 [10.1109/TGRS.2012.2230008]

L172, is there any technology breakthrough insight for the expectation of having a lidar on a GEO orbit at ~35000km? I wonder if there is powerful enough for being deployed at such a high altitude.

We agree here that the technology we are suggesting may be beyond the 20-year horizon. We have changed the statement.

**Minor comments**

Abstract:

As the Terra was launched in 1999, it is rather imprecise to say "Twenty years ago" (24 years ago by now) at the beginning of the abstract.

Better to say "retrieving parameters of aerosol loading, ..." than "retrieving information..."

"concept of applying algorithms to a single sensor will no longer exist". The "concept" always exists but may not be used in generating operational products, but the concept will remain valuable in teaching the principle of aerosol RS.

The list of factors contributing to the revolution of aerosol remote sensing in the past two decades may also include major motivation factors in the applications of aerosol remote sensing products for climate change, environment, and public health studies, among others.

We made wording changes in the abstract that incorporate Prof. Li's comments.

L146-1147, to my knowledge, HSRL has been dropped out from the AOS mission due to descoping

Yes, NASA cut it after we sent in the Opinion. We now mention the descoping of the HSRL from AOS in the paragraph about lidars.

L154, "foresee" instead of "see" would be better

ok

L180, how is the 15-60X information estimated? And how independent are they, noting that neighboring spectral channels are highly correlated.

3 polarization states X 5 along track view angles as per SPEXone as an example minimum number = 15X

3 polarization states X 20 along track view angles as per HARP2 for a max number – 60X

All for the same wavelength. We are assuming the same number of wavelengths for a single view radiometer. We are not looking for spectral information content here, but for size of the data acquisition. We have changed 'information' to 'data'. True information content will be less.

L223, sounds awkward expression "......become complete"

Changed to, "the LUT methods will be permanently jettisoned,"

L255, cite the applications of using the AI for these tasks, e.g. cloud identification (Wei et al., 2020), surface reflectance determination (Su et al. 2020)

Wei, J., W. Huang, Z. Li, L. Sun, X. Zhu, Q. Yuan, L. Liu, and M. C. Cribb, 2020: Cloud detection for Landsat imagery by combining the random forest and superpixels extracted via energy-driven sampling segmentation approaches, Remote Sens. Environ., 248, 112005, doi:10.1016/j.rse.2020.112005.

Su, T., I. Laszlo, Z. Li, J. Wei, and S. Kalluri, 2020: Refining aerosol optical depth retrievals over land by constructing the relationship of spectral surface reflectances through deep learning: application to Himawari-8, Remote Sens. Environ., 251, 112093, doi:10.1016/j.rse.2020.112093.

Ok. Thanks for the references.

L256, The statement "almost all ...physically-based..." sounds too sweeping.

We took out the "almost all"

L285, also state disadvantages: the quality of the data is highly variable that change with period, location, altitude.

We added a small paragraph listing what we see are the main disadvantages.

Fig. 1, Why not include MAIAC, another NASA official AOD product whose sum (together with DT&DB) reflects more thoroughly the total usage of the RS AOD product, a fairer comparison with the usage of the AOD assimilation products from MERRA2.

The problem is that MAIAC is a late comer to the table. There are two major references for MAIAC users: Lypustin et al. (2018) with 328 citations and Lyapustin et al. (2011) with 156 citations. We ended up using the 2018 one with the greater number of references. Because of the late start (2018 instead of 2013 for the Levy and Hsu papers) MAIAC gives an artificial acceleration to the MODIS aerosol citation time series. But in the end it didn't matter. The number of MODIS citations increased because of MAIAC but the temporal acceleration is not noticeable. The story is the same. MERRA use is accelerating while the use of the standard satellite products has plateaued.

We did change the figure to include MAIAC.

L307, "nano-sized cubesats…", I wonder if nano is the right phrase here, would "mini" be better?

Nano is the word they are using in the small sat community. We will keep it.

L340, "upgrade ….without loosing accuracy" seems illogic, how can an upgrade lead to loosing accuracy.

Prof. Li is correct.  It makes no sense. We took out "without losing accuracy". We can't remember what we were trying to say.

L374, "commercial networks will be everywhere" too exaggerating, no way they will be everywhere. I'd think they are mostly in populated areas, likely more so than in flagship networks.

Sure. Population centers of developed countries.  Made that change.

---

## Author Response (AR2)

Reviewer's comments in plain text.  **Our response in bold.**

As I mentioned in my earlier review of this paper, this is an opinion piece, and the reviewer need not agree with the authors, except on factual points. So, here again are a few thoughts from my perspective, for the authors to consider if they wish.

**Now that the reviewer has revealed has identity, we would like to personally thank Dr. Kahn for his two excellent thought-stimulating reviews of our Opinion.**

Abstract and Line 124: I'm wondering whether "hyperspectral" would always be the best choice if it comes at the expense of spatial resolution or coverage, or instrument cost. Similar thought regrading hyper-angle.

**We believe the abstract is accurate in the way it is presently stated,**

**"We anticipate technology that will replace today's standard multi-wavelength radiometers with hyperspectral and/or polarimetry all viewing in multiple angles."**

**Look at PACE.  There's a hyperspectral radiometer that tilts.  Sure to avoid glint, but it is not a big step to make something like that into a two-angle radiometer like the AATSR family. Complementing the radiometer are two multi-angle polarimeters.  The Abstract statement is and/or.  We believe that is accurate.**

**Line 124.  Yes.  We see Dr. Kahn's point and have modified the wording to avoid a reader from visualizing a super instrument.  New text,**

**"Future imaging polarimeters will improve upon the POLDER/PARASOL technology with better polarization accuracy, hyper spectral and hyper angle capabilities, decreases in pixel size, increase in the number of wavelengths with polarization and still allow for imaging. Super polarimeters with *all* of this technology in a single instrument are not necessary. Individual missions will tailor their instrument characteristics to meet particular mission needs, but these types of instrument advancement will be available for mission design.  "**

Abstract and Lines 508-509: I expect that applying algorithms to single sensors will probably still occur, in part because a lot can be retrieved from an instrument that combines the spectral range from the UV to the IR with multi-angle and polarization capabilities, and in part because the retrieval uncertainty will have to be estimated, and that will depend on each individual instrument's radiometric calibration uncertainty as well as the uncertainties in the assumptions that must be made to retrieve aerosol constraints, which will be different for instruments having different measurement capabilities. Further, the importance of error and uncertainty estimation, especially when multiple instruments having different capabilities contribute to a single retrieval, I think deserved more emphasis.

**Dr. Kahn keeps jumping on all of our provocative statements!**

**We have changed the wording in the abstract and at line 508 to read "in decline" rather than "extinct".**
**And we agree about the difficulty in applying uncertainty estimates to multi-sensor inversions. We have added the sentence,**

**"The caveat with multi-sensor inversions is the difficulty of determining the propagation of error and uncertainty from sensors of various capability and calibration."**

Abstract, and Lines 321-323, and it goes beyond what is stated in Lines 361-364: At least currently, "assimilation" has a specific meaning that you describe in Section 4, but it might not apply to every way in which measurements can be used to constrain models. Not all satellite datasets useful for constraining aerosol representation in models are well sampled spatially, are regularly sampled temporally, or have formal uncertainty estimates. For example, in addition to characterizing aerosol sources in models based on measurements, improving the parameterization of aerosol processes such as chemical evolution and aerosol-cloud interactions in models cannot be done via assimilation. (In case it is of interest, the issue of constraining aerosol modeling with measurements more generally is discussed, e.g., in Kahn et al., doi:10.1029/2022RG000796; also relevant for improving climate prediction and air quality forecasting (mentioned in Section 7).)

**The cyborg idea is another provocative concept! Dr. Kahn is correct in that satellite aerosol products have many uses besides assimilation. We have chosen to emphasize the cyborg idea as a way to provoke a conversation with our readers. Because of that we prefer to keep the Abstract, as is, and increase the discussion of satellite product interaction with models in the text. We have added the following to the final paragraph of Section 4.**

**"Using satellite-derived data sets as separate, independent data can improve parameterization of aerosol processes within models. Such processes include chemical transformation and aerosol-cloud interactions. Such independent constraints on models have important implications for climate prediction and air quality applications (Kahn et al., 2022)."**

Section 5. I know this goes beyond the scope of the current paper, but there are issues with space debris creating hazards for satellites (and space stations); at some point, actions might be taken to limit the number and type of material placed into orbit – hard to tell whether that would occur by 2043.

**True. I do know that at least in Europe all small sats now require propulsion in the design to enforce an efficient burn up during re-entry. But, it is beyond scope.**

Lines 528-529. As long as climate prediction matters, people will still be interested in aerosol radiative forcing. If the world is lucky enough not to have to worry about climate change by 2043, so much the better. However, at present that seems overly optimistic.

**Ok. Ok. We will modify our language again.**

**"Those interested in aerosol radiative forcing, will find numbers in Wm$^{-2}$ coming automatically out of some assimilation system."**

And Lines 522-523. If researchers are doing careful work, and their results depend on some estimate of aerosol amount, they will still likely use AOD, at least to test aspects of their results. AOD has the advantage of being easier to retrieve with confidence than aerosol quantities that also depends on the particle SSA or mass-extinction efficiency.

**Validation is a good point. We prefer to keep the leading sentence, as is, but have added validation as another use of AOD in the final sentence of the paragraph.**

**"The only interest in aerosol optical depth will be users working towards continuity with the old sensors for long-term trend analysis or as a critical first step in validating a retrieval."**

A few detailed suggestions:

Line 54. The following reference supersedes Martonchik et al. (2002): Martonchik, J.V., et al., 2009. Retrieval of Aerosol Properties over Land Using MISR Observations. In: Kokhanovsky, A.A. and G. de Leeuw, ed., Satellite Aerosol Remote Sensing Over Land. Springer, Berlin, pp.267-293. ISBN 978-3-540-69396-3.

**Thank you for the reference update. Implemented.**

Line 56. A better reference than Kahn et al. (1998) here would be: Kahn, R.A., P. Banerjee, and D. McDonald, 2001. The Sensitivity of Multiangle Imaging to Natural Mixtures of Aerosols Over Ocean, J. Geophys. Res.106, 18219-18238, doi: 10.1029/2000JD900497.

**Thank you again for the updated reference. Implemented.**

Line 166. Might be: "…lidars flying now or expected in the near future…"

**Implemented. Thank you.**

Table 1. You might use an asterisk to distinguish those missions that have launched successfully from those yet to be deployed.

Lines 270-271. Cost-function minimization does not require replacing the LUT; we actually use LUTs and apply cost-function minimization in MISR retrievals.

**The wording here was introduced after the last round of reviewer comments. It did not quite convey the meaning we intended. Yes, cost-function minimization can be applied to LUTs. We have made slight wording adjustments.**

**"Today, as information content increases and computer power grows to meet demand, LUTs are being supplemented by other methods to solve the radiative transfer equation. Techniques are being used to simultaneously retrieve multiple aerosol, gases, and surface parameters."**

Lines 434-437. AERONET indices of refraction, SSA, and particle sphericity variables are remote-sensing retrieval results that require many more assumptions and entail much greater uncertainties than the spectral AOD. At best, these are column-effective values that do not capture the distinct particle properties in the atmospheric column under the frequent circumstance that multiple aerosol modes are present. These limitations are often ignored by users of the data. For example, we can *compare* MISR-retrieved SSA with AERONET values, but we cannot validate the MISR results with AERONET data; the AERONET results are often no better, and sometimes not as good, as those from MISR. (This leads to the need for systematic in situ measurements if we are to improve key aspects of climate simulation and prediction, and also to the desirability of incorporating modeling in the retrieval process, e.g., to help constrain aerosol type by identifying the likely aerosol sources.)

**Dr. Kahn is correct. We knew this but didn't take manuscript space to write it. We have added text.**

**"The AERONET retrieval that produces reliable aerosol characterization in addition to AOD, including particle size distribution, single scattering albedo, complex refractive indices, and non-sphericity, will become a more important asset over time as aerosol satellite remote sensing advances in information content, requiring evaluation of satellite-retrieved products that include a wide range of aerosol parameters. However, we note that while the uncertainty in AERONET AOD is sufficiently small to offer true validation to a collocated satellite AOD product, the uncertainty of AERONET inversion products (particle size distribution, single scattering albedo etc.) is not. The AERONET inversion products are themselves subject to assumptions and caveats, and may have greater error than certain satellite parameters. Still, the inversion products provide a vital service in the evaluation of satellite products, providing a standardized framework of high quality, widely distributed aerosol characterization for nearly immediate comparisons with satellite products."**

Line 469. Why only in developed countries?

**We took those words out.**